# Multifaceted barriers associated with clinical breast examination in sub-Saharan Africa: A multilevel analytical approach

**Castro Ayebeng[1,2], Joshua Okyere[2,3]\*, Christiana Okantey[4], Isaac Yeboah Addo[5,6]**

**1** School of Demography, Australian National University, Canberra, Australian Capital Territory, Australia, **2** Department of Population and Health, University of Cape Coast, Cape Coast, Ghana, **3** School of Human and Health Sciences, University of Huddersfield, Huddersfield, England, United Kingdom, **4** Department of Adult Health, School of Nursing and Midwifery, University of Cape Coast, Cape Coast, Ghana, **5** Concord Clinical School, University of Sydney, Sydney, Australia, **6** Centre for Social Research in Health, The University of New South Wales, Sydney, Australia

\* joshuaokyere54@gmail.com, joshua.okyere@hud.ac.uk

## Abstract

### Objectives

Clinical breast examination (CBE) open the pathway to early detection and diagnosis of breast cancer. This study examined barriers to CBE uptake in seven sub-Saharan African (SSA) countries.

### Methods

Data from the most current Demographic and Health Surveys of Burkina Faso, Cote d'Ivoire, Ghana, and Kenya Mozambique, Senegal and Tanzania was used. A weighted sample size of 65,486 women aged 25–49 years was used to estimate the pooled prevalence of CBE. We employed a multilevel logistic regression modelling technique, with results presented in adjusted odds ratios (aOR) along with a 95% confidence interval (CI).

### Results

The pooled prevalence of CBE uptake in the studied SSA countries is low at 19.2% [95%CI: 18.5–19.8]. Screening uptake was significantly low among women reporting difficulty in getting permission (aOR = 0.88, 95% CI: 0.82–0.95), and distance (aOR = 0.95, 95% CI: 0.89–0.99), as well as those who reported financial constraints (aOR = 0.92, 95% CI: 0.88–0.97), as barriers to access healthcare facilities. However, surprisingly, women who faced travel-alone barriers were 1.19 times (95%CI: 1.10–1.28) more likely to utilise CBE than those who did not face this barrier.

### Conclusions

We conclude that barriers such as difficulties in obtaining permission, long distances to healthcare facilities, and financial constraints significantly reduce the likelihood of women undergoing CBE. The study underscores a need to improve access to healthcare facilities.

**Data Availability Statement:** The datasets generated and/or analysed during the current study are available in the Measure DHS repository: http:

www.//dhsprogram.com/data/available-datasets.
cfm.

**Funding:** The author(s) received no specific
funding for this work.

**Competing interests:** The authors have declared
that no competing interests exist.

**Abbreviations:** aOR, adjusted odds ratios; CBE,
Clinical breast examination; DHS, Demographic
and Health Survey; SSA, sub-Saharan Africa.

Practically, this can be achieved by expanding mobile health services and integrating CBE into primary healthcare will help overcome distance-related challenges. Additionally, targeted outreach and transportation initiatives are necessary to support women facing travel barriers.

## Background

Breast cancer is a significant public health challenge globally, particularly in low- and middle-income countries (LMICs) where late-stage diagnosis and limited access to treatment contribute to high mortality rates [1, 2]. Breast cancer screening is crucial for early detection, and allows for less aggressive treatment, increased survival rates, and a better quality of life for patients [3, 4]. Breast cancer screening can also assist in detecting precancerous changes in the body, enabling preventive measures to reduce the risk of developing invasive breast cancer, and as a result, reducing treatment costs as treating early-stage cancer is generally less expensive than treating advanced-stage cancer [5–7]. Beyond individual benefits, breast cancer screening contributes to public health efforts by reducing the overall burden of breast cancer and improving population health outcomes, providing women with proactive control over their breast health [8].

A variety of methods can be employed in breast cancer screening with the objective of early detection of the disease. One common approach is mammography, which involves X-ray imaging of the breast tissues [9]. Mammography is widely known to be particularly effective for women over 40 years and is considered the gold standard for detecting breast cancer at early, more treatable stages [10, 11]. Breast self-examination (BSE) is another screening method that involves women examining their own breasts regularly to detect any changes or abnormalities [12]. While BSE is simple and cost-effective, its effectiveness in reducing breast cancer mortality is debated, and it may lead to unnecessary anxieties if the result is a false positive [13, 14]. Other methods, such as breast ultrasound and magnetic resonance imaging (MRI), are sometimes used in specific cases, such as for women with dense breast tissues or those at high risk of breast cancer [15, 16]. Essentially, the choice of screening method depends on various factors, including age, risk factors, and resource availability; and a combination of methods may be used for optimal screening outcomes [17]. Another significant screening method warranting attention is the clinical breast examination (CBE). CBE is a cost-effective screening method recommended for early detection of breast cancer, especially in settings with limited resources and often involves a healthcare professional manually examining the breasts for lumps or other abnormalities [18, 19]. Nevertheless, the utility of CBE in high-income countries such as the United States is currently subject to debate due to the widespread presence and advancements in mammography technology in such countries [20].

In Sub-Saharan Africa (SSA) where resources for breast screening are constrained despite the significant risk and burden of breast cancer, CBE is regarded as a crucial screening method for breast cancer prevention and early intervention. A recent cancer mapping study involving all 54 African countries revealed that breast cancer is the most predominant malignancy among females in the continent, surpassing 34 other identified cancer types, with a total of 186,598 new cases and 85,787 fatalities recorded solely in the year 2020 [21]. However, the prevalence of CBE utilisation and the barriers to its uptake in SSA, remain poorly understood. Thus, there is a significant dearth of studies in SSA that have focused on examining the prevalence of CBE uptake and identifying barriers to its utilisation. The lack of research can be

attributed to various factors, such as resource constraints, with limited funding, infrastructure, and trained personnel restricting large-scale studies [22–25]. Additionally, the extensive prioritisation of infectious disease control, such as malaria and HIV over non-communicable diseases like breast cancer diverts research efforts and funding away from breast cancer screening behaviour studies [26–30]. Moreover, the lack of comprehensive health data in cancer registries and health information systems in several African countries poses additional challenges for designing population-based studies on CBE [24, 31]. By pooling data from seven SSA countries (Burkina Faso, Cote d'Ivoire, Ghana, Kenya, Mozambique, Senegal and Tanzania) where variables related to CBE are available in the Demographic and Health Surveys (DHS) dataset, this study aimed to examine the status of CBE utilisation and identified common barriers that hinder its uptake. Findings from this study are essential for policymakers, healthcare providers, and researchers working to improve breast cancer screening and early detection programs in SSA.

## Methods

### Data source and design

This is a secondary cross-sectional analysis from seven SSA countries (Burkina Faso (2021), Cote d'Ivoire (2021), Ghana (2022), Kenya (2022), Mozambique (2022–23), Senegal (2023) and Tanzania (2022)) where variables related to CBE are available in the most recent (within the last five years) Demographic and Health Surveys (DHS) dataset. The DHS data is a nationally representative data derived from two-stage stratified sampling procedure in selecting research participants of households in low- and middle-income countries (LMICs). In the first phase, clusters/enumeration areas (EAs) are selected, guided by a sample frame developed during the preceding census of the respective countries, while in the second stage, a sample of households is drawn from each selected EA. Within selected households, women between the ages of 15 and 49 years were asked to provide information regarding their households, personal details, and their children, as well as some health screening behaviours such as screening for clinical breast cancer which is the outcome of interest in this study. The surveys employed similar data collection methodologies and comparable questionnaires, enabling cross-country comparisons [32]. This study analysed a weighted sample of 65,486 women aged 25–49 years from the individual recode file (IR file). We excluded women aged 15–24 years due to the low risk of breast cancer in this population [33]. Detailed determination of the final sample size for this study is specified in Fig 1.

### Variables and measures

**Outcome variable.** Clinical breast examination (CBE) is the outcome of interest in this study. This was derived from the question, "Have you ever had your breast examined by a healthcare provider? The responses were dichotomised as "yes = 1" or "no = 0".

**Exposure variables.** Informed by previous studies [34–36], and in alignment with the DHS data four barriers were used as the main exposure variables in this study. These include difficulties in getting permission to access healthcare, distance to the nearest health facility, getting money needed for treatment, and travelling alone. All these barrier variables have a binary response as "a big problem = 1" and "not a big problem = 0" as captured in the DHS dataset. In addition to these barriers, other individual and community-level factors were identified and adjusted for in the analyses based on empirical literature [25, 34–36].

The individual-level factors were age (25–29, 30–34, 35–39, 40–44, and 45–49), level of education (no formal education, basic, secondary and higher), employment status (working or not working), religion (Christianity, Islam, and other), visit health facility in the last 12 months

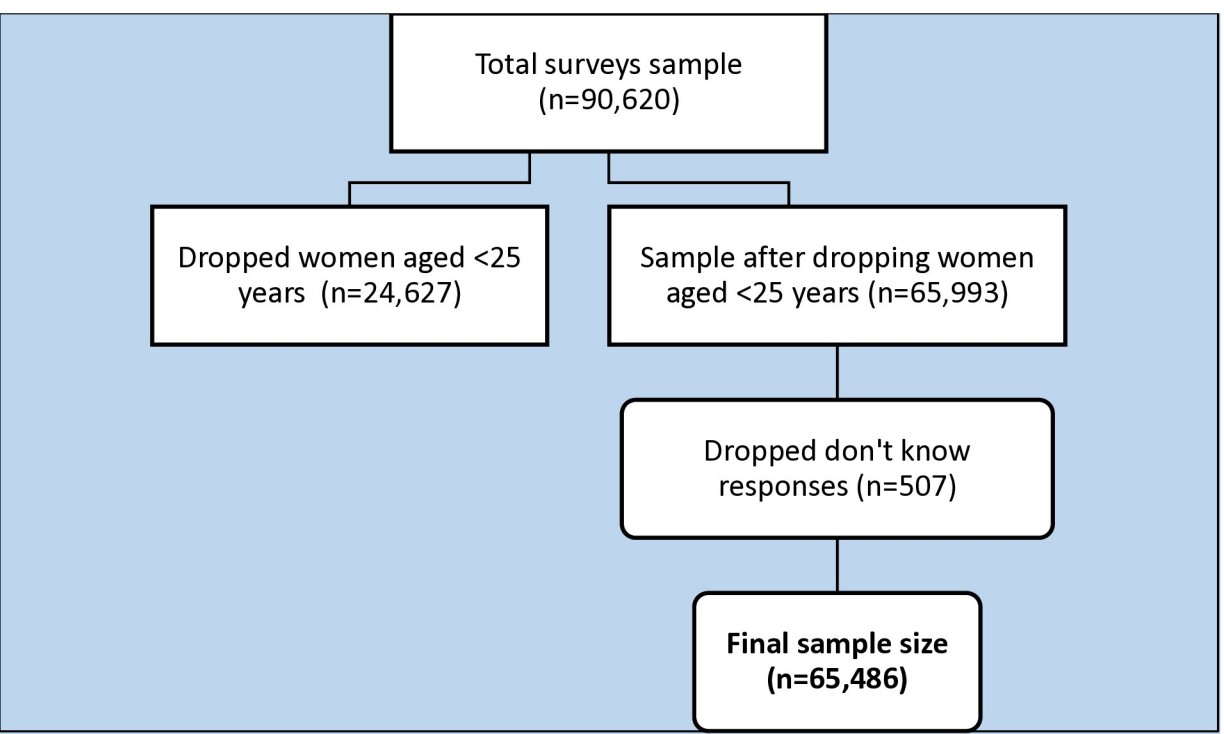

**Fig 1. Flow chart of sample determination.**

(yes or no), wealth index (poorest, poorer, middle, richer and richest), contraceptive usage (not using, modern contraceptive, and traditional method), and media exposure. Media exposure is a composite variable derived from three sets of variables: frequency of reading newspapers or magazines; frequency of listening to the radio; and frequency of watching television. The responses were categorised as (not at all = 0, less than once a week = 1, at least once a week = 2, and almost every day = 3). We used the 'egen' command in STATA to generate the composite variable where those who did not listen to the radio, watch TV or read newspaper/magazine at all were categorised as "no media exposure", while the remaining responses were categorised as having media exposure. The community-level factors were type of residence (rural or urban) and country of residence.

## Analytical procedure

We conducted a descriptive analysis of the percentage of women screened for CBE across seven countries, presenting distributions across identified barriers with chi-square tests for associations. Due to the hierarchical data structure where women were nested within a cluster, we used multilevel logistic regression models. Four models were constructed: the null model, Model I with the four main barriers (permission, money, distance, traveling alone), Model II adjusted for individual factors (age, education, employment, religion, wealth, health facility visits, contraceptive use, media exposure), and Model III including community-level factors (residence, country). Model IV adjusted for the effects of the individual, and community-level factors on the association between barrier variables and the outcome of interest. Odds ratios with 95% confidence intervals were presented, and multicollinearity was assessed with a variance inflation factor (VIF) mean score of 4.49. Analyses were performed using Stata 17,

applying sample weights. The complex sampling design of the DHS survey was accounted for by using the "*svyset*" command in Stata (which takes into account the primary sampling units (PSU) and the stratification of the clusters).

To apply sample weights in a pooled data analysis, the standard weight variable for the individual recode file (v005) was first de-normalized as follows: v005 × (total female population 15–49 in the country)/(total number of women 15–49 interviewed in the survey) and then re-normalized so that the pooled sample average is 1. This was significant because, according to the DHS sampling and household listing manual, the normalised weight is not valid for pooled data, even data pooled for women and men in the same survey, because the normalisation factor is country and sex-specific [37].

### Fixed and random effect estimates

The analysis incorporated both fixed and random effects. In the fixed effect analysis, we examined a set of variables encompassing individual and contextual factors. The goodness of fit of the mixed effect models was assessed using the Log-likelihood (LL) test, Akaike's Information Criterion (AIC) and Bayesian Information Criterion (BIC). The model characterised by the lowest value of the information criterion was chosen as the best model in the analysis. The random effect analysis was aimed to assess variations between clusters (EAs). We calculated two essential metrics to quantify these variations: Intra-class correlation coefficient (ICC) and the proportional change in variance (PCV). The ICC gauges the extent of variation within clusters, specifically among individuals within the same cluster. It was computed using the formula:

$$ICC = [V_A/(V_A + \pi2/3)] = V_A/(V_A + 3.29).$$

Here, $V_A$ represents the estimated variance in each model [38].

To assess the overall variation attributed to individual and contextual factors in each model, we utilised the proportional change in variance (PCV), calculated as:

$$PCV = (V_A - V_B)/V_A.$$

In this equation, $V_A$ is the variance of the initial model, and $V_B$ is the variance of the model with additional terms [38]. Our presentation of the results was guided by the Strengthening the Reporting of Observational Studies in Epidemiology (STROBE) guidelines (S1 File) [39].

### Ethical approval and consent to participate

Ethical clearance was not required for our study because we used a publicly available DHS dataset. However, the DHS reports that both written and verbal informed consent were obtained from all participants. We accessed the datasets from the DHS Program after completing the necessary registration and obtaining approval for their use. We adhered to all ethical guidelines relevant to the use of the secondary dataset in research publications.

## Results

### Prevalence of CBE

Overall, the pooled prevalence of CBE across all the studied countries was 19.2% (95%CI: 18.5–19.8) (Table 1). However, the prevalence of CBE uptake varied across countries with the highest uptake being reported in Burkina Faso 31.1% (95%CI: 29.1–33.1) and the least from Tanzania 7.3% (95%CI: 6.5–8.2).

**Table 1. Prevalence of CBE among women by countries.**

| Country | Survey year | Weighted sample | Percentage of women who attended CBE [95%CI] |
|---|---|---|---|
| Burkina Faso | 2021 | 10,550 | 31.1[29.1–33.1] |
| Cote d'Ivoire | 2021 | 8,959 | 17.0[15.3–18.9] |
| Ghana | 2022 | 9,639 | 22.5[21.0–24.2] |
| Kenya | 2022 | 10,494 | 18.5[17.4–19.6] |
| Mozambique | 2022–23 | 6,977 | 11.2[9.9–12.6] |
| Senegal | 2023 | 9,455 | 22.9[21.1–24.9] |
| Tanzania | 2022 | 9,413 | 7.3[6.5–8.2] |
| **All countries** | | **65,486** | **19.2[18.5–19.8]** |

[#]All percentages are calculated in row percentages

### Distribution of clinical breast examination across identified barriers by country

In Burkina Faso, Cote d'Ivoire, Ghana and Mozambique, difficulty in getting permission to visit health facilities emerged as a significant barrier, with only 5.5% (p = 0.014) utilising CBE among women who considered it a big problem in Mozambique. Difficulties in getting money needed for treatment also posed a significant barrier across all countries (p < 0.001), with substantially lower proportions in Mozambique (5.6%) and Tanzania (5.5%) undergoing CBE among those who found it challenging. Distance to a health facility was another significant barrier (p < 0.001), with a significantly lower percentage of women who consider it a big problem in all countries screening for CBE, particularly in Tanzania (4.9%) and Mozambique (5.4%) (Table 2).

Additionally, travelling alone was identified as an important barrier in Cote d'Ivoire, Ghana, Kenya, Mozambique, and Tanzania, with a notably lower proportion of the respondents in Mozambique (8%) and Tanzania (6.2%) screening for CBE among those who considered travelling alone as a challenge. Overall, the presence of at least one of the barriers significantly influenced the percentage of women undergoing CBE across all countries (p < 0.001), with significantly lower proportions undergoing screening in Mozambique (6.9%) and Tanzania (5.6%).

### Proportion of CBE utilisation among women by selected covariates

Table 3 shows the distribution of CBE utilization by selected covariates. CBE utilization was high among women aged 35–39 years 20.6%(95%CI: 19.6–21.7), those with higher education 38.5%(95%CI: 36.4–40.6), those currently employed 20.7%(95%CI: 20.0–21.5), had visited a health facility in the last 12 months 22.9%(95%CI: 22.1–23.7), were using modern 21.7%(95%CI: 20.9–22.7) or traditional contraceptive methods 24.6%(95%CI: 22.6–26.8), had media exposure 21.9%(95%CI: 21.2–22.6), and resided in the urban setting 24.5%(95%CI: 23.5–25.5).

### Barriers associated with CBE uptake among women in seven SSA countries

Women reporting difficulty in getting permission (aOR = 0.88, 95% CI: 0.82–0.95), and distance (aOR = 0.95, 95% CI: 0.89–0.99), as well as those who reported financial constraints (aOR = 0.92, 95%CI: 0.88–0.97) to access healthcare facilities, had lower odds of CBE uptake. However, surprisingly, women who faced travel-alone barriers were 1.19 times (95%CI: 1.10–1.28) more likely to utilise CBE compared to those who did not face this barrier.

**Table 2. Distribution of clinical breast examination across identified barriers by country.**

| Barriers | Clinical Breast Examination (CBE) | | | | | | | | | | | | | |
|---|---|---|---|---|---|---|---|---|---|---|---|---|---|---|
| | Burkina Faso | | Cote d'Ivoire | | Ghana | | Kenya | | Mozambique | | Senegal | | Tanzania | |
| | [n(%)] | p-value | [n(%)] | p-value | [n(%)] | p-value | [n(%)] | p-value | [n(%)] | p-value | [n(%)] | p-value | [n(%)] | p-value |
| **Permission to visit health facility** | | <0.001 | | <0.001 | | <0.001 | | 0.093 | | 0.014 | | 0.574 | | 0.557 |
| Big problem | 365 (24.0) | | 412 (12.4) | | 149 (17.1) | | 91 (17.9) | | 33 (5.5) | | 566 (23.3) | | 40 (6.4) | |
| Not a big problem | 2,922 (32.2) | | 1,150 (19.7) | | 1,994 (23.1) | | 1,845 (18.5) | | 764 (11.6) | | 1,566 (22.8) | | 643 (7.3) | |
| **Money needed for treatment** | | <0.001 | | <0.001 | | <0.001 | | <0.001 | | <0.001 | | 0.006 | | <0.001 |
| Big problem | 1,900 (27.9) | | 851 (14.5) | | 762 (17.4) | | 822 (16.3) | | 137 (5.6) | | 1,365 (22.4) | | 191 (5.5) | |
| Not a big problem | 1,386 (36.9) | | 711 (21.6) | | 1,381 (26.9) | | 1,114 (20.5) | | 660 (14.2) | | 767 (24.1) | | 491 (8.4) | |
| **Distance to a health facility** | | <0.001 | | <0.001 | | <0.001 | | <0.001 | | <0.001 | | 0.774 | | <0.001 |
| Big problem | 1,062 (26.8) | | 541 (13.7) | | 343 (15.8) | | 403 (15.5) | | 152 (5.4) | | 801 (24.0) | | 135 (4.9) | |
| Not a big problem | 2,225 (33.6) | | 1,021 (19.5) | | 1,800 (24.8) | | 1,532 (19.5) | | 644 (15.0) | | 1,332 (22.4) | | 548 (8.3) | |
| **Travel alone** | | 0.451 | | <0.001 | | <0.001 | | 0.009 | | 0.002 | | 0.284 | | 0.014 |
| Big problem | 514 (30.5) | | 290 (11.9) | | 260 (18.8) | | 146 (17.0) | | 53 (8.0) | | 542 (25.5) | | 88 (6.2) | |
| Not a big problem | 2,773 (31.2) | | 1,272 (18.9) | | 1,883 (23.2) | | 2,352 (18.6) | | 744 (11.6) | | 1,591 (22.2) | | 595 (7.5) | |
| **At least one barrier** | | <0.001 | | <0.001 | | <0.001 | | <0.001 | | <0.001 | | 0.006 | | <0.001 |
| Big problem | 2,122 (28.8) | | 952 (14.7) | | 924 (18.1) | | 943 (16.8) | | 239 (6.9) | | 1,467 (22.3) | | 263 (5.6) | |
| Not a big problem | 1,165 (36.4) | | 611 (22.7) | | 1,219 (27.8) | | 992 (20.5) | | 557 (15.4) | | 665 (24.4) | | 419 (9.0) | |

[#]All percentages are calculated in row percentages

Higher screening odds were observed among women aged 35–39 years (aOR = 1.31, 95% CI: 1.22–1.40), Muslim (aOR = 1.56, 95%CI: 1.34–1.80) and Christians (aOR = 1.81, 95%CI: 1.57–2.09), and those from the richest households (aOR = 2.01, 95%CI: 1.81–2.23). The odds of undergoing CBE were low among women with no formal education (aOR = 0.37, 95%CI: 0.33–0.41) or having only primary education (aOR = 0.43, 95%CI: 0.39–0.47), those who had no visitation to a health facility in the last 12 months (aOR = 0.62, 95%CI: 0.59–0.66), and women with no media exposure (aOR = 0.75, 95%CI: 0.70–0.80). Women who were using modern or traditional methods of contraceptives were 1.20 (95%CI: 1.14–1.27) and 1.32(95% CI: 1.19–1.47) times more likely to undergo CBE than those who were not using any form of contraceptives.

Additionally, community-level factors such as residence and country of residence exhibited substantial effect, with women resident in rural areas (aOR = 0.88, 95%CI: 0.83–0.94), as well as all countries showing significantly reduced odds of CBE uptake compared to the reference country (Burkina Faso).

**Random effects results.** The random-effect models demonstrated that community-level variance slightly increased from 0.152 in the null model to 0.196 in the full model, indicating

**Table 3. Proportion of CBE utilisation among women by selected covariates.**

| Covariates | Women who screened for CBE | | p-value |
|---|---|---|---|
| | **Frequency (n)** | **Percent (%)[95%CI]** | |
| **Age** | | | <0.001 |
| 25–29 years | 3,033 | 17.6[16.8–18.5] | |
| 30–34 years | 2,967 | 19.5[18.5–20.5] | |
| 35–39 years | 2,876 | 20.6[19.6–21.7] | |
| 40–44 years | 2,129 | 19.7[18.7–20.8] | |
| 45–49 years | 1,550 | 18.7[17.5–19.9] | |
| **Education** | | | <0.001 |
| No education | 4,245 | 16.8[15.9–17.8] | |
| Primary | 2,630 | 13.5[12.8–14.3] | |
| Secondary | 3,671 | 23.6[22.5–24.7] | |
| higher | 2,008 | 38.5[36.4–40.6] | |
| **Occupational status** | | | <0.001 |
| Working | 8,849 | 20.7[20.0–21.5] | |
| Not working | 3,707 | 16.2[15.3–17.2] | |
| **Marital status** | | | <0.001 |
| Never married | 1,059 | 20.1[18.5–21.7] | |
| Married | 10,127 | 19.5[18.8–20.2] | |
| Formerly married | 1,369 | 16.8[15.6–18.0] | |
| **Religion** | | | <0.001 |
| Muslim | 5,195 | 21.9[20.8–23.3] | |
| Christian | 6,457 | 21.5[20.7–22.3] | |
| Other | 227 | 9.6[8.1–11.2] | |
| **Wealth index** | | | <0.001 |
| Poorest | 1,167 | 10.5[9.6–11.5] | |
| Poorer | 1,603 | 14.0[13.0–15.1] | |
| Middle | 2,045 | 16.4[15.3–17.5] | |
| Richer | 2,831 | 20.0[18.9–21.2] | |
| Richest | 4,909 | 30.0[28.8–31.3] | |
| **Visit health facility in last 12 months** | | | <0.001 |
| No | 3,339 | 13.3[12.6–14.0] | |
| Yes | 9,217 | 22.9[22.1–23.7] | |
| **Contraceptive usage** | | | <0.001 |
| Not using | 7,434 | 17.5[16.9–18.3] | |
| Modern contraceptive | 4,354 | 21.7[20.9–22.7] | |
| Traditional method | 768 | 24.6[22.6–26.8] | |
| **Media exposure within a week** | | | <0.001 |
| Exposed | 10,968 | 21.9[21.2–22.6] | |
| unexposed | 1,587 | 10.3[9.5–11.2] | |
| **Type of residence** | | | <0.001 |
| Urban | 7,129 | 24.5[23.5–25.5] | |
| Rural | 5,426 | 14.9[14.1–15.7] | |

#All percentages are calculated in row percentages

some variability in the outcome due to differences between communities. The proportional change in variance (PCV) indicates that 18.1% of the variation between communities is explained by the full model compared to the null model. The intra-class correlation coefficient (ICC) remains fairly consistent, ranging from 4.1% to 4.8%, suggesting that about 4–5% of the total variability in the outcome can be attributed to differences between communities across the models. This highlights the importance of accounting for community-level factors in understanding the variation in the outcome (see Table 4).

**Model fit statistics.** The model fitness statistics evaluate how well the multilevel logistic regression model fits the data. We employed the Loglikelihood (LL) test, Akaike's Information Criterion (AIC), and Bayesian Information Criterion (BIC) for comparing different models. Lower AIC and BIC values indicate a better fit. Notably, in our study, Model 4 exhibited the lowest AIC score (50439.6) and BIC (50707.65), indicating that it offers a better fit to the data compared to the other models.

## Discussion

There is universal consensus that secondary prevention practices such as CBE open the pathway to early detection and diagnosis of breast cancer, and in the long run, help to reduce breast cancer-related mortalities [5, 40]. Yet, many women face barriers when they attempt to seek CBE. Against this background, we investigated the uptake of and barriers associated in SSA. Our findings show a low uptake of CBE among women in SSA 19.2%. The estimated prevalence of CBE uptake is supported by existing literature from SSA [35] that argues that there is a low uptake of CBE services. Additionally, Addo et al. [34] corroborate this, reporting a similarly low uptake of 16.3% among SSA women. Although these figures underscore a significant gap, there is a slight improvement noted compared to Ba et al.'s findings [36], which reported a 12% uptake. This suggests that efforts to raise awareness, improve accessibility, and enhance healthcare infrastructure may be yielding some positive results in increasing CBE uptake in SSA. This low uptake of CBE services underscores the pressing need for continued interventions and initiatives aimed at further enhancing CBE utilisation across SSA.

Our study revealed that women who had difficulty with getting permission to seek healthcare were significantly less likely to undergo screening for cervical cancer. Similar findings have been reported in a systematic review [41] that found challenges in gaining permission from husbands/partners as a major barrier to breast cancer screening. The result is also consistent with Mahajan et al. [42] findings that show lack of permission from partners as a barrier to screening uptake. Onyenwenyi and Mchunu [43] argue that in SSA countries like Nigeria, male partners do not consent for women to undergo CBE, and this stems from the fear that the woman would be examined by a male health professional. It is also imperative to understand that in many SSA countries, women may be expected to seek permission from male partners before accessing healthcare services, reflecting broader patriarchal structures that prioritize male authority. This undermines the woman's autonomy to freely make decisions about their health and practicing preventive behaviors.

Distance also emerged as a significant barrier to the uptake of CBE services in SSA. This finding resonates with Addo et al. [34] who found a strong inverse association between women's consideration of distance to the healthcare facility as a problem and their uptake of CBE services. The result is corroborated by Srinath et al. [41] who argued that perceived long distance to screening centres tend to be a major hurdle to women's utilisation of CBE services. Beyond the logistical challenges posed by geographical distance [44], we propose that women's decision-making regarding CBE uptake in SSA may be influenced by a complex cost-benefit analysis. In regions like SSA where resources are scarce and competing demands for time and

**Table 4. Multilevel logistic regression analysis of barriers associated with CBE in Seven SSA countries.**

| Explanatory variables | Null model (0) | Model I | Model II | Model III | Model IV (full model) |
|---|---|---|---|---|---|
| | | aOR(95%CI) | aOR(95%CI) | aOR(95%CI) | aOR(95%CI) |
| *Fixed effect*: | | | | | |
| **Permission** | | | | | |
| Not a problem | | 1.00 | 1.00 | 1.00 | 1.00 |
| A big problem | | 0.98[0.91–1.05] | 0.88***[0.81–0.94] | 0.90**[0.84–0.97] | 0.88**[0.82–0.95] |
| **Money** | | | | | |
| Not a problem | | 1.00 | 1.00 | 1.00 | 1.00 |
| A big problem | | 0.89***[0.85–0.94] | 1.03[0.98–1.08] | 0.75***[0.75–0.79] | 0.92**[0.88–0.97] |
| **Distance** | | | | | |
| Not a problem | | 1.00 | 1.00 | 1.00 | 1.00 |
| A big problem | | 0.70***[0.67–0.74] | 0.90***[0.84–0.95] | 0.86***[0.81–0.91] | 0.95*[0.89–0.99] |
| **Travel alone** | | | | | |
| Not a problem | | 1.00 | 1.00 | 1.00 | 1.00 |
| A big problem | | 1.13***[1.06–1.21] | 1.21***[1.13–1.30] | 1.14***[1.07–1.23] | 1.19***[1.10–1.28] |
| **Age (years)** | | | | | |
| 25–29 | | | 1.00 | | 1.00 |
| 30–34 | | | 1.15***[1.08–1.22] | | 1.17***[1.10–1.25] |
| 35–39 | | | 1.25***[1.17–1.34] | | 1.31***[1.22–1.40] |
| 40–44 | | | 1.24***[1.16–1.33] | | 1.29***[1.20–1.39] |
| 45–49 | | | 1.20***[1.11–1.30] | | 1.26***[1.17–1.37] |
| **Education** | | | | | |
| No education | | | 0.55***[0.50–0.60] | | 0.37***[0.33–0.41] |
| Primary | | | 0.46***[0.43–0.51] | | 0.43***[0.39–0.47] |
| Secondary | | | 0.68***[0.62–0.73] | | 0.59***[0.54–0.64] |
| Higher | | | 1.00 | | 1.00 |
| **Occupational status** | | | | | |
| Working | | | 1.00 | | 1.00 |
| Not working | | | 0.84***[0.80–0.89] | | 0.91***[0.86–0.95] |
| **Religion** | | | | | |
| Other | | | 1.00 | | 1.00 |
| Muslim | | | 2.15***[1.87–2.47] | | 1.56***[1.34–1.80] |
| Christian | | | 1.63***[1.41–1.87] | | 1.81***[1.57–2.09] |
| **Visit health facility in last 12 months** | | | | | |
| Yes | | | 1.00 | | 1.00 |
| No | | | 0.58***[0.55–0.61] | | 0.62***[0.59–0.66] |
| **Wealth index** | | | | | |
| Poorest | | | 1.00 | | 1.00 |
| Poorer | | | 1.23***[1.13–1.33] | | 1.16***[1.07–1.26] |
| Middle | | | 1.40***[1.29–1.51] | | 1.28***[1.17–1.39] |
| Richer | | | 1.62***[1.49–1.76] | | 1.42***[1.30–1.56] |
| Richest | | | 2.55***[2.34–2.77] | | 2.01***[1.81–2.23] |
| **Contraceptive usage** | | | | | |
| Not using | | | 1.00 | | 1.00 |
| Modern contraceptive | | | 1.20***[1.15–1.26] | | 1.20***[1.14–1.27] |
| Traditional method | | | 1.31***[1.18–1.46] | | 1.32***[1.19–1.47] |
| **Media exposure within a week** | | | | | |
| Exposed | | | 1.00 | | 1.00 |

*(Continued)*

**Table 4.** (Continued)

| Explanatory variables | Null model (0) | Model I | Model II | Model III | Model IV (full model) |
|---|---|---|---|---|---|
| | | aOR(95%CI) | aOR(95%CI) | aOR(95%CI) | aOR(95%CI) |
| Unexposed | | | 0.71***[0.67–0.76] | | 0.75***[0.70–0.80] |
| *Community-level factors* | | | | | |
| **Type of residence** | | | | | |
| Urban | | | | 1.00 | 1.00 |
| Rural | | | | 0.53***[0.50–0.51] | 0.88***[0.83–0.94] |
| **Country** | | | | | |
| Burkina Faso | | | | 1.00 | 1.00 |
| Cote d'Ivoire | | | | 0.32***[0.30–0.35] | 0.37***[0.33–0.39] |
| Ghana | | | | 0.45***[0.42–0.48] | 0.38***[0.34–0.40] |
| Kenya | | | | 0.33***[0.30–0.35] | 0.24***[0.20–0.24] |
| Mozambique | | | | 0.27***[0.25–0.29] | 0.23***[0.21–0.25] |
| Senegal | | | | 0.55***[0.55–0.63] | 0.66***[0.61–0.72] |
| Tanzania | | | | 0.13***[0.12–0.15] | 0.***[0.78–0.91] |
| *Random-effect model* | | | | | |
| Community variance (SE) | 0.152(0.01) | 0.146(0.01) | 0.142(0.01) | 0.166(0.02) | 0.196(0.01) |
| PCV (%) | 1.00 | -.1.15 | 7.0 | 16.9 | 18.1 |
| ICC (%) | 4.4 | 4.2 | 4.1 | 4.6 | 4.7 |
| *Model fit statistics* | | | | | |
| Log-likelihood | -30475.116 | -30335.514 | -25923.555 | -28759.601 | -25189.799 |
| AIC | 60954.23 | 60683.03 | 51895.11 | 57537.2 | 50439.6 |
| BIC | 60972.41 | 60737.57 | 52109.55 | 57619.01 | 50707.65 |

1.00: reference category; aOR: adjusted odds ratio; 95%CI: 95% confidence interval; ICC: Intraclass correlation; PCV: Proportional change in variance; AIC: Akaike information criterion; BIC: Bayesian information criterion

*p<0.05,

**p<0.01,

***p<0.001

financial resources are prevalent, women must weigh the perceived costs of traveling long distances to healthcare facilities against the anticipated benefits of undergoing screening [45]. This calculus extends beyond mere transportation expenses to encompass opportunity costs, such as lost wages from taking time off work or the potential need for childcare during the screening process. As such, the perceived inconvenience of travelling long distance to access CBE services may serve as a deterrent, particularly for women who face economic hardship or have limited access to information about the importance of early screening [41, 45].

Consistent with extant literature from Ethiopia [46], Tanzania [47], and Ghana [48], our findings showed that women who had a big problem with getting money for healthcare were less likely to undergo screening for CBE. This is expected as CBE in the SSA countries included in this study comes at a cost. This means that women who lack the financial capacity or struggle to get financial support will find it challenging to afford the cost of screening as well as other ancillary costs such as the cost of transportation from their homes to the healthcare facility. An unexpected finding was the association between travelling alone and screening uptake. We found that women who considered travelling alone as a big problem were more likely to undergo CBE. This is surprising as it challenges the notion that autonomy is critical for improving women's uptake of CBE services. Further research is required to fully comprehend the nuances surrounding this unexpected observation.

Beyond the main hypothesis of testing the extent to which barriers to healthcare access influence CBE uptake, we identified significant associations across the covariates. Increasing age, having higher educational attainment, being employed, having visited the healthcare facility, being exposed to the media, and being in the richest wealth index were associated with higher screening uptake–a result that aligns with a plethora of studies [34–36]. Additionally, women who used some sort of contraception were more likely to undergo screening. This may be explained from the perspective that women who use contraceptives are health conscious and prioritise preventive health measures.

## Implications for policy

Considering the insights from this study, it is imperative to invest in empowering women in SSA to become more assertive and autonomous in their healthcare decision-making. Practical initiatives can include equipping women with skills to engage in economic livelihoods, drawing from successful empowerment programs in similar settings. For example, initiatives such as microfinance and vocational training programs in SSA have demonstrated significant outcomes in enhancing women's economic independence and decision-making power [49]. Addressing the financial barriers and boosting women's confidence to independently seek CBE services align with the Sustainable Development Goals (SDGs), particularly SDG 5 (Gender Equality) and SDG 3 (Good Health and Well-being). Furthermore, to address distance barriers to screening uptake, the health departments, and Ministries of Health in SSA countries should consider integrating CBE services at the primary healthcare level. This approach would bring screening services closer to communities, making them more accessible and reducing logistical challenges for women seeking care.

## Strengths and limitations

This study relied on recent DHS data (2021/2023). Thus, our findings a reflection of the current status quo of SSA countries. Also, the large sample size coupled with the multi-stage sampling technique of the DHS ensures that the sample analyzed in this study is representative at the national and population level. The use of multi-level regression also adds to the strength of the study as it allows us to account for the different levels of factors that influence breast cancer screening uptake. Nonetheless, no causal relationship can be established due to the cross-sectional nature of the DHS. Given that we relied on secondary data, we were restricted to only variables available in the dataset. This means that other barriers (e.g., cultural norms and expectations, knowledge/information barriers, availability of facilities that provide CBE services, affordability of CBE services, healthcare provider attitudes, etc.) could not be accounted for in our model. Moreover, the study included only seven SSA countries that have available DHS datasets with a variable measuring CBE in the individual recode file (IR file), which makes us limited to extrapolating our results to all SSA countries. Similarly, our findings are limited to women of reproductive age; thus, the findings may not represent the situation in older women (50 years and older).

## Conclusion

We conclude that barriers such as difficulties in obtaining permission, long distances to healthcare facilities, and financial constraints significantly reduce the likelihood of women undergoing CBE. The study underscores a need to improve access to healthcare facilities. Practically, this can be achieved by expanding mobile health services and integrating CBE into primary healthcare will help overcome distance-related challenges. Additionally, targeted outreach and transportation initiatives are necessary to support women facing travel barriers.

## Supporting information

**S1 File. STROBE checklist.**
(DOCX)

## Acknowledgments

We acknowledge the Measure DHS for granting us free access to the dataset used in this study.

## Author Contributions

**Conceptualization:** Castro Ayebeng, Joshua Okyere, Christiana Okantey, Isaac Yeboah Addo.

**Data curation:** Castro Ayebeng, Joshua Okyere.

**Formal analysis:** Castro Ayebeng.

**Methodology:** Castro Ayebeng, Joshua Okyere, Christiana Okantey, Isaac Yeboah Addo.

**Validation:** Joshua Okyere, Isaac Yeboah Addo.

**Writing – original draft:** Castro Ayebeng, Joshua Okyere, Christiana Okantey, Isaac Yeboah Addo.

**Writing – review & editing:** Castro Ayebeng, Joshua Okyere, Christiana Okantey, Isaac Yeboah Addo.

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
