## [Decision Letter · Decision Letter 0]

15 Sep 2024

PONE-D-24-26002Multifaceted barriers impacting clinical breast examination in sub-Saharan Africa: A multilevel analytical approachPLOS ONE

Dear Dr. Okyere,

Thank you for submitting your manuscript to PLOS ONE. After careful consideration, we feel that it has merit but does not fully meet PLOS ONE’s publication criteria as it currently stands. Therefore, we invite you to submit a revised version of the manuscript that addresses the points raised during the review process.

Please note the editor also serve as an additional reviewers and provide the comments below.

Thank you for the opportunity to review this manuscript, which attempts to disentangle the barriers affecting women's access to clinical breast examinations in Sub-Saharan African (SSA) countries, using recent DHS data.

I have one major concern. As outlined in the DHS documentation, specifically the 'Guide to DHS Statistics,' DHS data are generated using a complex sampling design that includes both cluster and stratified sampling, along with sampling probabilities. When using DHS data involving multiple levels (e.g., household, census enumeration areas, national level), which I believe applies to the data used in this manuscript, it is essential to incorporate not only sample weights but also the primary sampling unit (PSU) variable and stratification variable in the analyses. However, these important analytical details are not mentioned. I assume they were not incorporated into the analyses presented in the manuscript. (In response to this comment, the authors may need more time than usual for revisions. If this is the case, please contact PLOS ONE to request an extension.) 

It’s unclear why only Burkina Faso, Cote d’Ivoire, Ghana and Kenya were selected if the study findings are intended to be generalized across all SSA countries. The manuscript states that these four countries were selected "based on the defined timeframe, context, and availability of variables of interest" (Page 3). What exactly is the defined timeframe? What specific variables of interest were unavailable in the datasets from other SSA countries? How to justify that the study findings from those 4 countries are generalizable to the broader SSA region?

The manuscript also lacks sufficient details about the studied barriers, i.e. permission, money, distance, and traveling alone. What does ‘permission to access healthcare’ really refer to? Which survey datasets were used to obtain each of these variables? The justification provided ("Informed by previous studies [34, 35, 36]") is too vague and shallow. In the revision, please create a supplemental file detailing which specific DHS data files were used for acquiring these key study variables, as well as other variables. Additionally, I suggest providing the programming codes used for the modeling analysis to allow readers to reproduce the results.  

Some expressions in the manuscript are vague or confusing. For example, “… to implement comprehensive strategies aimed at empowering women.” The phrase "empowering women" seems to suggest more than just reducing the barriers discussed in the paper.

Please add more information to the tables, such as the sample size (N) for columns, whether percentages are row or column percentages, add in a table footnote the specific analytical approaches used to generate the results in each table.

We look forward to receiving your revised manuscript.

Kind regards,

Ruofei Du, PhD

Academic Editor

PLOS ONE

Journal requirements: 1. When submitting your revision, we need you to address these additional requirements. Please ensure that your manuscript meets PLOS ONE's style requirements, including those for file naming. The PLOS ONE style templates can be found at https://journals.plos.org/plosone/s/file?id=wjVg/PLOSOne_formatting_sample_main_body.pdf and https://journals.plos.org/plosone/s/file?id=ba62/PLOSOne_formatting_sample_title_authors_affiliations.pdf. 2. Your ethics statement should only appear in the Methods section of your manuscript. If your ethics statement is written in any section besides the Methods, please move it to the Methods section and delete it from any other section. Please ensure that your ethics statement is included in your manuscript, as the ethics statement entered into the online submission form will not be published alongside your manuscript.  3. Please upload a copy of Supplementary File 1 to which you refer in your text on page 22. Please amend the file type to 'Supporting Information'. If the Supplementary file is no longer to be included as part of the submission please remove all reference to it within the text.

Reviewers' comments:

Reviewer's Responses to Questions

**Comments to the Author**

1. Is the manuscript technically sound, and do the data support the conclusions?

Reviewer #1: Yes

Reviewer #2: Partly

2. Has the statistical analysis been performed appropriately and rigorously? 

Reviewer #1: Yes

Reviewer #2: I Don't Know

3. Have the authors made all data underlying the findings in their manuscript fully available?

Reviewer #1: Yes

Reviewer #2: Yes

4. Is the manuscript presented in an intelligible fashion and written in standard English?

Reviewer #1: Yes

Reviewer #2: Yes

5. Review Comments to the Author

Reviewer #1: Thank you for the opportunity to review the manuscript titled “Multifaceted barriers impacting clinical breast examination in sub-Saharan Africa: A multilevel analytical approach.”

This is well written manuscript and good work was done by the authors

I have some suggestions for the authors to consider

Results

• Page 9: “However, women who faced travel alone barrier were 1.13 times more likely to utilise CBE compared to those who were not.” The authors should provide the confidence interval in support of the odds

• “Women who were using modern or traditional methods of contraceptives were 1.18 and 1.31 times more likely to undergo CBE than those who were not using any form of contraceptives.” The authors should provide the confidence interval in support of the odds when reporting the odds in a signal phrase or as the integral of the text. Odds alone might not be enough in making some comparison of the outcome of variables, confidence interval plays a major role in that so try and provide that as a support to the odds

Discussion

• First Paragraph: “Despite this modest improvement, it is crucial to acknowledge that the current uptake rates remain far below optimal levels.” When a comparison statement is made like this, it will be very crucial to state the comparators, therefore, I will urge the authors to state the optimal level of uptake

•

Reviewer #2: Major

1. Potential Bias Due to Dropped Missing Values (n = 15,255)

The manuscript mentions that a substantial number of observations (n = 15,255) were excluded due to missing values. The exclusion of such a large portion of the sample may introduce significant bias into the study results. The authors should discuss the potential impact of these missing values on the findings. Specifically, they should consider whether the missing data is random or if it is systematically related to certain characteristics (e.g., socioeconomic status, location, education level) that could influence the outcome (CBE uptake). If the missing data is not completely random, it could skew the results, leading to biased estimates of the association between the exposure variables and CBE uptake.

2. Potential Bias Introduced by Dichotomized Measurements

The study utilizes dichotomized measurements for several key variables (e.g., barriers such as permission, financial constraints, distance, and traveling alone). While dichotomization simplifies the analysis, it may lead to a loss of information and potentially introduce bias. Dichotomizing variables can obscure important variations within the data and create artificial cut-off points that may not accurately reflect the true distribution or relationship of the variables in the population. For instance, treating 'distance' or 'financial constraints' as binary variables might oversimplify complex barriers that exist on a continuum. The authors should discuss these limitations in the manuscript and consider alternative approaches, such as using the original continuous or ordinal scales, which may provide a more nuanced understanding of the relationships between the variables and CBE uptake.

3. Table 2 Fallacy: Misinterpretation of Exposure and Outcome Relationships

The use of a single model in Table 4 to estimate the effects of four exposures (permission, financial constraints, distance, and traveling alone) on one outcome (CBE uptake) with the same set of confounders adjusted for all exposures presents a potential issue known as the "Table 2 fallacy" (Westreich & Greenland, 2013). This approach may lead to misinterpretation of the associations due to the assumption that the same set of covariates confounds all exposures equally. Each exposure-outcome relationship should ideally be considered separately with its own directed acyclic graph (DAG) to identify the minimal sufficient adjustment set for each specific exposure. The authors are encouraged to revise the analytical strategy to reflect this. By identifying and adjusting for the correct set of covariates for each exposure separately, the authors can provide more accurate effect estimates and interpretations, thus strengthening the validity of their findings.

6. PLOS authors have the option to publish the peer review history of their article (what does this mean?). If published, this will include your full peer review and any attached files.

Reviewer #1: **Yes: **Robert Kokou Dowou

Reviewer #2: No

---

## [Author Response · Author response to Decision Letter 0]

25 Sep 2024

PONE-D-24-26002

Multifaceted barriers impacting clinical breast examination in sub-Saharan Africa: A multilevel analytical approach

Dear Editor,

We are grateful to you and the reviewers for your comments on our paper. We would also take this opportunity to thank the reviewers for finding merit in this paper and suggesting some revisions. We have taken notice of all the comments raised by the reviewers and have responded accordingly as follows. Please note that the reviewers' comments are in black whereas our responses are in red, and have been highlighted accordingly in the revised manuscript. 

 I have one major concern. As outlined in the DHS documentation, specifically the 'Guide to DHS Statistics,' DHS data are generated using a complex sampling design that includes both cluster and stratified sampling, along with sampling probabilities. When using DHS data involving multiple levels (e.g., household, census enumeration areas, national level), which I believe applies to the data used in this manuscript, it is essential to incorporate not only sample weights but also the primary sampling unit (PSU) variable and stratification variable in the analyses. However, these important analytical details are not mentioned. I assume they were not incorporated into the analyses presented in the manuscript. (In response to this comment, the authors may need more time than usual for revisions. If this is the case, please contact PLOS ONE to request an extension.) 

Response: We agree with you. This was taken into consideration in analyzing our study which has been specified in the methods “The complex sampling design of the DHS survey was accounted for by using the "svyset" command in Stata (which takes into account the primary sampling units (PSU) and the stratification of the clusters). Kindly refer to pp.7. Thank you.

It’s unclear why only Burkina Faso, Cote d’Ivoire, Ghana and Kenya were selected if the study findings are intended to be generalized across all SSA countries. The manuscript states that these four countries were selected "based on the defined timeframe, context, and availability of variables of interest" (Page 3). What exactly is the defined timeframe? What specific variables of interest were unavailable in the datasets from other SSA countries? 

Response: Thank you for this comment. At the initial stage of analyzing the study, only these four countries had a variable that measures Clinical Breast Examination (CBE). However, an additional three countries (Mozambique, Senegal and Tanzania) have published a recent DHS dataset that measures CBE. Hence the analysis has been revised to include these countries making seven. Overall, these seven countries were selected based on the availability of the outcome variable in the most recent (within the last five years) Demographic and Health Surveys (DHS) dataset.

How to justify that the study findings from those 4 countries are generalizable to the broader SSA region?

Response: Thank you for your invaluable contribution. This limitation has been acknowledged which reads “Moreover, the study included only seven SSA countries that have available DHS datasets with a variable measuring CBE in the individual recode file (IR file), which makes us limited to extrapolating our results to all SSA countries”.

The manuscript also lacks sufficient details about the studied barriers, i.e. permission, money, distance, and traveling alone. What does ‘permission to access healthcare’ really refer to? Which survey datasets were used to obtain each of these variables? The justification provided ("Informed by previous studies [34, 35, 36]") is too vague and shallow. 

Response: Thank you for this comment. These four barrier variables are captured by the DHS dataset measuring difficulties accessing healthcare facilities. These include difficulty in getting permission to access healthcare, distance to the nearest health facility, getting money needed for treatment, and travelling alone. All these barrier variables have a binary response as “a big problem=1” and “not a big problem=0” as captured in the DHS dataset”. Hence, our decision to select these exposures was informed by their availability in the dataset, as well as their inclusion in existing studies.

In the revision, please create a supplemental file detailing which specific DHS data files were used for acquiring these key study variables, as well as other variables. Additionally, I suggest providing the programming codes used for the modeling analysis to allow readers to reproduce the results. 

Response: The study used the individual recode data (IR file) of the DHS dataset of respective countries with an available variable that measures CBE. We have ensured that a detailed description of the analytical procedure is provided in the method which is enough for others to replicate the study, instead of sharing our programming codes. Thank you.

Some expressions in the manuscript are vague or confusing. For example, “… to implement comprehensive strategies aimed at empowering women.” The phrase "empowering women" seems to suggest more than just reducing the barriers discussed in the paper.

Response: Thank you for the comment. We have now removed this sentence. 

Please add more information to the tables, such as the sample size (N) for columns, whether percentages are row or column percentages, add in a table footnote the specific analytical approaches used to generate the results in each table.

Response: Thank you. A footnote has been provided under Table 1-3 indicating that all percentages are calculated in row percentages.

Response: Thank you. This concern has been addressed. Kindly refer to pp.8.

3. Please upload a copy of Supplementary File 1 to which you refer in your text on page 22. Please amend the file type to 'Supporting Information'. If the Supplementary file is no longer to be included as part of the submission please remove all reference to it within the text.

Response: We have uploaded the supplementary file. 

Reviewer #1: 

Thank you for the opportunity to review the manuscript titled “Multifaceted barriers impacting clinical breast examination in sub-Saharan Africa: A multilevel analytical approach.”

This is well written manuscript and good work was done by the authors

I have some suggestions for the authors to consider

Results

• Page 9: “However, women who faced travel alone barrier were 1.13 times more likely to utilise CBE compared to those who were not.” The authors should provide the confidence interval in support of the odds

Response: Thank you. The confidence interval has been added “However, women who faced travel-alone barriers were 1.19 times (95%CI: 1.10-1.28) more likely to utilise CBE compared to those who did not face travel-alone as a barrier” Please refer to pp.10.

• “Women who were using modern or traditional methods of contraceptives were 1.18 and 1.31 times more likely to undergo CBE than those who were not using any form of contraceptives.” The authors should provide the confidence interval in support of the odds when reporting the odds in a signal phrase or as the integral of the text. Odds alone might not be enough in making some comparison of the outcome of variables, confidence interval plays a major role in that so try and provide that as a support to the odds

Response: This has been addressed “Women who were using modern or traditional methods of contraceptives were 1.20 (95%CI: 1.14-1.27) and 1.32(95%CI: 1.19-1.47) times more likely to undergo CBE than those who were not using any form of contraceptives” Please refer to pp.10.

Discussion

• First Paragraph: “Despite this modest improvement, it is crucial to acknowledge that the current uptake rates remain far below optimal levels.” When a comparison statement is made like this, it will be very crucial to state the comparators, therefore, I will urge the authors to state the optimal level of uptake

Response: Thank you for the comment. We have now addressed this. 

Reviewer #2: Major

1. Potential Bias Due to Dropped Missing Values (n = 15,255).

The manuscript mentions that a substantial number of observations (n = 15,255) were excluded due to missing values. The exclusion of such a large portion of the sample may introduce significant bias into the study results. The authors should discuss the potential impact of these missing values on the findings. Specifically, they should consider whether the missing data is random or if it is systematically related to certain characteristics (e.g., socioeconomic status, location, education level) that could influence the outcome (CBE uptake). If the missing data is not completely random, it could skew the results, leading to biased estimates of the association between the exposure variables and CBE uptake.

Response: Thank you for drawing attention to this. The error has been corrected. Kindly refer to the flow chart (Figure 1) showing the sample size determination. We observed no missing values in the selected variables in the study, with the exception of “507 don’t know responses” which were dropped from the analysis.

2. Potential Bias Introduced by Dichotomized Measurements. The study utilizes dichotomized measurements for several key variables (e.g., barriers such as permission, financial constraints, distance, and traveling alone). While dichotomization simplifies the analysis, it may lead to a loss of information and potentially introduce bias. Dichotomizing variables can obscure important variations within the data and create artificial cut-off points that may not accurately reflect the true distribution or relationship of the variables in the population. For instance, treating 'distance' or 'financial constraints' as binary variables might oversimplify complex barriers that exist on a continuum. The authors should discuss these limitations in the manuscript and consider alternative approaches, such as using the original continuous or ordinal scales, which may provide a more nuanced understanding of the relationships between the variables and CBE uptake.

Response: Thank you for this insight. However, kindly be informed that these four variables were captured as a binary response category in the DHS dataset, thus whether the respondent is finding it difficult to access healthcare facilities or not based on these barriers.

3. Table 2 Fallacy: Misinterpretation of Exposure and Outcome Relationships

The use of a single model in Table 4 to estimate the effects of four exposures (permission, financial constraints, distance, and traveling alone) on one outcome (CBE uptake) with the same set of confounders adjusted for all exposures presents a potential issue known as the "Table 2 fallacy" (Westreich & Greenland, 2013). This approach may lead to misinterpretation of the associations due to the assumption that the same set of covariates confounds all exposures equally. Each exposure-outcome relationship should ideally be considered separately with its own directed acyclic graph (DAG) to identify the minimal sufficient adjustment set for each specific exposure. The authors are encouraged to revise the analytical strategy to reflect this. By identifying and adjusting for the correct set of covariates for each exposure separately, the authors can provide more accurate effect estimates and interpretations, thus strengthening the validity of their findings.

Response: Thank you for your comment. However, table 4 does not suffer from "Table 2 fallacy." It presents results from multilevel logistic regression models, showing how the adjusted odds ratios (aORs) of various barriers and community-level factors evolve across different models (from the null model to the full model). In effects, it highlights how each variable affects the outcome of interest as more explanatory variables are added. The progressive models and the model fit statistics (AIC, BIC, Log-likelihood) are included, allowing for a robust assessment of the impact of the variables. Therefore, the analysis considers the whole model evolution and outcomes, which mitigates the risk of “Table 2 fallacy”. Besides, the interpretation of the results is based on the full model.

---

## [Decision Letter · Decision Letter 1]

29 Oct 2024

PONE-D-24-26002R1Multifaceted barriers impacting clinical breast examination in sub-Saharan Africa: A multilevel analytical approachPLOS ONE

Dear Dr. Okyere,

Thank you for submitting your manuscript to PLOS ONE. After careful consideration, we feel that it has merit but does not fully meet PLOS ONE’s publication criteria as it currently stands. Therefore, we invite you to submit a revised version of the manuscript that addresses the points raised during the review process.

(Please note: The Academic Editor is also serving as a reviewer for this submission.)

Thanks for addressing the initial comments, particularly by clarifying analytical details. However, there remain major concerns in the analyses:

1. The DHS data provides sample weights within each country, but does not include guidance for combining results across multiple countries. This suggests the analyses would be conducted separately by country, as shown in Tables 1 and 2, with the 'All countries' row removed. Logistic regressions, currently presented in Table 4, should also be fitted separately by country.

2. In Table 4, it appears that Model IV is intended as the final working model. Presenting results from the other models that were not selected may be confusing, which echoes the comments from the other reviewer. It might also be helpful to emphasize they are adjusted association effects instead of casual effects per se.

3. In the current model, 'Travel alone' does not show a statistically significant association with CBE uptake, while it does show significance in the marginal associations in Table 2 for most countries. This discrepancy may result from correlations among covariates in the final model. It’s understandable that 'Distance' and 'Travel alone' might be correlated. Then it’s a question whether the model selection procedure has completed. Consider performing backward or stepwise selection to refine the model, potentially retaining only one of 'Distance' or 'Travel alone' in the final model for correct finding and interpretation.

Additional minor comments:

1. The authors included random effects in the models to account for cluster effects. Could you clarify what these clusters represent? Do they represent regions within a country? So are they the random effect nested in each country? 

2. There are hypothesis tests for random effects. They provide direct evidence of including random effects or not in model approaches.

3. On page 9, in the first sentence of the first paragraph, the country name should be "Mozambique" instead of "Cote d'Ivoire" in the context: "with only 5.5% (p=0.014)...a big problem in Cote d'Ivoire." Please check the text thoroughly for accuracy in all details.

We look forward to receiving your revised manuscript.

Kind regards,

Ruofei Du, PhD

Academic Editor

PLOS ONE

Reviewers' comments:

Reviewer's Responses to Questions

**Comments to the Author**

1. If the authors have adequately addressed your comments raised in a previous round of review and you feel that this manuscript is now acceptable for publication, you may indicate that here to bypass the “Comments to the Author” section, enter your conflict of interest statement in the “Confidential to Editor” section, and submit your "Accept" recommendation.

Reviewer #2: (No Response)

2. Is the manuscript technically sound, and do the data support the conclusions?

Reviewer #2: Partly

3. Has the statistical analysis been performed appropriately and rigorously? 

Reviewer #2: Yes

4. Have the authors made all data underlying the findings in their manuscript fully available?

Reviewer #2: Yes

5. Is the manuscript presented in an intelligible fashion and written in standard English?

Reviewer #2: Yes

6. Review Comments to the Author

Reviewer #2: Table 2 Fallacy: Misinterpretation of Exposure and Outcome Relationships

The manuscript presents four models in Table 4, which aim to estimate the effects of different factors on Clinical Breast Examination (CBE) uptake:

• Model 1: CBE uptake ~ Permission + Money + Distance + Travel alone

• Model 2: CBE uptake ~ Age + Education + Occupational status + Religion + Health facility visit (last 12 months) + Wealth index + Contraceptive usage + Media exposure (within a week)

• Model 3: CBE uptake ~ Type of residence + Country

• Model 4: CBE uptake ~ All variables

The authors estimate the causal effects of four specific factors on CBE uptake (Permission, Money, Distance, and Travel alone) by calculating adjusted odds ratios (aORs) in the full model (Model 4). While Model 4 uses criteria such as AIC, BIC, and Log-likelihood for model selection, it implicitly assumes that each of these four main factors is confounded by all other factors in the model, including other variables from Models 1, 2, and 3. This approach may lead to what is commonly known as the "Table 2 fallacy."

The "Table 2 fallacy," introduced by Westreich and Greenland (2013), occurs when multiple effect estimates are presented from a single model and mistakenly interpreted as causal effects. It involves assuming that each variable's relationship with the outcome is equally interpretable as causal, which can be misleading, especially when variables have distinct roles or relationships with the outcome.

To improve the analysis and avoid this Table 2 fallacy, I recommend the following steps to justify the confounding adjustment set:

1. Separate Tables for Each Exposure: Present each exposure's effect estimates and potential confounders in individual tables to reduce confusion.

2. Use of Causal Diagrams (DAGs): Employ directed acyclic graphs (DAGs) to clarify the causal relationships and identify which variables should be adjusted for in each model. Analyze each exposure variable separately, using a unique DAG for each, to determine the minimal sufficient adjustment set. This ensures that each model includes only the confounders relevant to that specific exposure-outcome relationship.

3. Tailored Confounder Sets: Refine the confounder set for each exposure variable to prevent over-adjustment or adjustment for irrelevant variables. Additionally, consider the following adjustments:

By taking these steps, you can better address the potential for misinterpretation and provide more robust conclusions on the factors affecting CBE uptake.

7. PLOS authors have the option to publish the peer review history of their article (what does this mean?). If published, this will include your full peer review and any attached files.

Reviewer #2: No

---

## [Author Response · Author response to Decision Letter 1]

3 Nov 2024

PONE-D-24-26002R1

Multifaceted barriers impacting clinical breast examination in sub-Saharan Africa: A multilevel analytical approach

PLOS ONE

 01 November, 2024

Dear Editor,

On behalf of the coauthors of Manuscript ID PONE-D-24-26002R1, I thank you and the reviewers for reviewing our manuscript, “Multifaceted barriers impacting clinical breast examination in sub-Saharan Africa: A multilevel analytical approach”. We have addressed each point raised by the reviewers below and tracked any corresponding changes in the manuscript

Thanks for addressing the initial comments, particularly by clarifying analytical details. However, there remain major concerns in the analyses:

1. The DHS data provides sample weights within each country but does not include guidance for combining results across multiple countries. This suggests the analyses would be conducted separately by country, as shown in Tables 1 and 2, with the 'All countries' row removed. Logistic regressions, currently presented in Table 4, should also be fitted separately by country.

Response: Thank you for your contribution. Please be informed that pooling DHS data from multiple countries is a common and robust research practice aligned with DHS guidelines on sampling and weighting procedures across countries. This approach also helps account for the effect of country-specific variations on the association under study. Detailed procedures for pooling DHS data for multi-country analysis can be found in the DHS Sampling and Household Listing Manual (pp. 26–29), available at https://dhsprogram.com/pubs/pdf/DHSM4/DHS6_Sampling_Manual_Sept2012_DHSM4.pdf

2. In Table 4, it appears that Model IV is intended as the final working model. Presenting results from the other models that were not selected may be confusing, which echoes the comments from the other reviewer. It might also be helpful to emphasize they are adjusted association effects instead of casual effects per se.

Response: Thank you for the insight. Presenting the results for each model, along with the final model, enables readers to better understand how individual and contextual factors contribute to the associations examined in the study. We have clarified that, given the study's cross-sectional design, the models show associations rather than causality; this limitation is also discussed in the limitations section.

3. In the current model, 'Travel alone' does not show a statistically significant association with CBE uptake, while it does show significance in the marginal associations in Table 2 for most countries. This discrepancy may result from correlations among covariates in the final model. It’s understandable that 'Distance' and 'Travel alone' might be correlated. Then it’s a question whether the model selection procedure has completed. Consider performing backward or stepwise selection to refine the model, potentially retaining only one of 'Distance' or 'Travel alone' in the final model for correct finding and interpretation.

Response: Thank you. Please note that we assessed multicollinearity among the main predictors, as well as the individual and contextual variables, using the variance inflation factor (VIF) before fitting the regression models. The assessment yielded a mean VIF of 4.49, indicating no significant collinearity issues. Additionally, we would like to highlight that in the current model, "Travel alone" is statistically significant, with an adjusted odds ratio (aOR)=1.19; 95%CI: 1.10–1.28.

Additional minor comments:

1. The authors included random effects in the models to account for cluster effects. Could you clarify what these clusters represent? Do they represent regions within a country? So are they the random effect nested in each country? 

Response: In our analysis, clusters represent enumeration areas (EAs) as per the DHS data structure. These clusters are indeed nested within each country and account for intracluster correlations, addressing potential biases from the hierarchical data structure. 

2. There are hypothesis tests for random effects. They provide direct evidence of including random effects or not in model approaches.

Response: In this study, random effects were included to account for the hierarchical structure of the data, where individual women were nested within clusters (enumeration areas (EAs)). To evaluate the significance of these random effects, we used the Intra-Class Correlation Coefficient (ICC) and Proportional Change in Variance (PCV). We appreciate your invaluable contribution.

3. On page 9, in the first sentence of the first paragraph, the country name should be "Mozambique" instead of "Cote d'Ivoire" in the context: "with only 5.5% (p=0.014)...a big problem in Cote d'Ivoire." Please check the text thoroughly for accuracy in all details.

Response: This error has been corrected. It reads “In Burkina Faso, Cote d’Ivoire, Ghana and Mozambique, difficulty in getting permission to visit health facilities emerged as a significant barrier, with only 5.5% (p=0.014) utilising CBE among women who considered it a big problem in Mozambique”. Thank you.

Reviewer #2: Table 2 Fallacy: Misinterpretation of Exposure and Outcome Relationships

The manuscript presents four models in Table 4, which aim to estimate the effects of different factors on Clinical Breast Examination (CBE) uptake:

• Model 1: CBE uptake ~ Permission + Money + Distance + Travel alone

• Model 2: CBE uptake ~ Age + Education + Occupational status + Religion + Health facility visit (last 12 months) + Wealth index + Contraceptive usage + Media exposure (within a week)

• Model 3: CBE uptake ~ Type of residence + Country

• Model 4: CBE uptake ~ All variables

The authors estimate the causal effects of four specific factors on CBE uptake (Permission, Money, Distance, and Travel alone) by calculating adjusted odds ratios (aORs) in the full model (Model 4). While Model 4 uses criteria such as AIC, BIC, and Log-likelihood for model selection, it implicitly assumes that each of these four main factors is confounded by all other factors in the model, including other variables from Models 1, 2, and 3. This approach may lead to what is commonly known as the "Table 2 fallacy."

The "Table 2 fallacy," introduced by Westreich and Greenland (2013), occurs when multiple effect estimates are presented from a single model and mistakenly interpreted as causal effects. It involves assuming that each variable's relationship with the outcome is equally interpretable as causal, which can be misleading, especially when variables have distinct roles or relationships with the outcome.

To improve the analysis and avoid this Table 2 fallacy, I recommend the following steps to justify the confounding adjustment set:

1. Separate Tables for Each Exposure: Present each exposure's effect estimates and potential confounders in individual tables to reduce confusion.

2. Use of Causal Diagrams (DAGs): Employ directed acyclic graphs (DAGs) to clarify the causal relationships and identify which variables should be adjusted for in each model. Analyze each exposure variable separately, using a unique DAG for each, to determine the minimal sufficient adjustment set. This ensures that each model includes only the confounders relevant to that specific exposure-outcome relationship.

3. Tailored Confounder Sets: Refine the confounder set for each exposure variable to prevent over-adjustment or adjustment for irrelevant variables. Additionally, consider the following adjustments:

By taking these steps, you can better address the potential for misinterpretation and provide more robust conclusions on the factors affecting CBE uptake.

Response: Thank you for your insightful contribution to this study. First and foremost, please note that this study relied on cross-sectional data, hence in effect, it does not establish any causality but rather an association between barrier variables (Permission + Money + Distance + Travel alone) and CBE uptake accounting for both individual and contextual level factors using multilevel logistic regression model instead of the traditional logistic regression model due to the hierarchical structure of the data, where individual women were nested within clusters (enumeration areas (EAs)). We would like to reemphasize that Table 4’s presentation, with Models I-IV, does not fall under the "Table 2 fallacy" as described by Westreich and Greenland, because it does not present each variable’s association with the outcome in isolation as if each relationship were causal. Instead, it shows a progression of models that incrementally adjust for various levels of covariates— both individual and contextual—reflecting an analytical framework to examine associations rather than causal inferences.

---

## [Decision Letter · Decision Letter 2]

18 Dec 2024

Multifaceted barriers associated with clinical breast examination in sub-Saharan Africa: A multilevel analytical approach

PONE-D-24-26002R2

Dear Dr. Okyere,

We’re pleased to inform you that your manuscript has been judged scientifically suitable for publication and will be formally accepted for publication once it meets all outstanding technical requirements.

Kind regards,

Ruofei Du, PhD

Academic Editor

PLOS ONE

Additional Editor Comments (optional):

Reviewers' comments:

Reviewer's Responses to Questions

**Comments to the Author**

1. If the authors have adequately addressed your comments raised in a previous round of review and you feel that this manuscript is now acceptable for publication, you may indicate that here to bypass the “Comments to the Author” section, enter your conflict of interest statement in the “Confidential to Editor” section, and submit your "Accept" recommendation.

Reviewer #2: All comments have been addressed

2. Is the manuscript technically sound, and do the data support the conclusions?

Reviewer #2: Yes

3. Has the statistical analysis been performed appropriately and rigorously? 

Reviewer #2: (No Response)

4. Have the authors made all data underlying the findings in their manuscript fully available?

Reviewer #2: Yes

5. Is the manuscript presented in an intelligible fashion and written in standard English?

Reviewer #2: Yes

6. Review Comments to the Author

Reviewer #2: (No Response)

7. PLOS authors have the option to publish the peer review history of their article (what does this mean?). If published, this will include your full peer review and any attached files.

Reviewer #2: No

---

## [Editor Report · Acceptance letter]

22 Dec 2024

PONE-D-24-26002R2 

PLOS ONE

Dear Dr. Okyere, 

I'm pleased to inform you that your manuscript has been deemed suitable for publication in PLOS ONE. Congratulations! Your manuscript is now being handed over to our production team.

Kind regards, 

on behalf of

Dr. Ruofei Du 

Academic Editor

PLOS ONE
